# An unscheduled switch to endocycles induces a reversible senescent arrest that impairs growth of the *Drosophila* wing disc

Yi-Ting Huang, Lauren L. Hesting, Brian R. Calvi *

Department of Biology, Simon Cancer Center, Indiana University, Bloomington, Indiana, United States of America

* bcalvi@iu.edu

**Data Availability Statement:** All relevant data are within the manuscript and its Supporting Information files.

## Abstract

A programmed developmental switch to G / S endocycles results in tissue growth through an increase in cell size. Unscheduled, induced endocycling cells (iECs) promote wound healing but also contribute to cancer. Much remains unknown, however, about how these iECs affect tissue growth. Using the *D. melanogaster* wing disc as model, we find that populations of iECs initially increase in size but then subsequently undergo a heterogenous arrest that causes severe tissue undergrowth. iECs acquired DNA damage and activated a Jun N-terminal kinase (JNK) pathway, but, unlike other stressed cells, were apoptosis-resistant and not eliminated from the epithelium. Instead, iECs entered a JNK-dependent and reversible senescent-like arrest. Senescent iECs promoted division of diploid neighbors, but this compensatory proliferation did not rescue tissue growth. Our study has uncovered unique attributes of iECs and their effects on tissue growth that have important implications for understanding their roles in wound healing and cancer.

## Author summary

The endocycle is an alternative growth program during which cells increase in size and repeatedly duplicate their DNA without dividing. The switch from cell division cycles to endocycles occurs normally during development of many tissues across plants and animals including humans. Cells can also switch to unscheduled endocycles in response to various conditions. Evidence suggests that this switch is beneficial for wound healing, but also can have pathological effects, most notably in cancer. Much remains unknown, however, about the regulation of these unscheduled endocycles and their impact on tissue growth. Using the *Drosophila* larval wing disc as model, we have found that unscheduled endocycles are limited in their growth by a specific type of senescent arrest that is mediated by a Jun Kinase stress pathway, which results in severe deleterious effects on tissue growth. We found that these arrested endocycling cells can go back to error prone divisions that compromise genomic DNA integrity, and can also promote the division of neighboring cells. Our study has revealed new inherent properties of unscheduled

**Funding:** This research was supported by the National Institutes of Health Grant NIH R01GM113107 and R35GM152255 to B.R.C. The funders had no role in study design, data collection and analysis, decision to publish, or preparation of the manuscript.

**Competing interests:** The authors have declared that no competing interests exist.

endocycling cells that impact tissue growth, with important implications for understanding their contribution to wound healing and cancer.

## Introduction

The regulation of tissue growth and homeostasis is not completely understood. Many tissues grow through a canonical mitotic cell cycle that increases cell number, with final tissue mass being determined by the total cell number and cell size. During the development of some tissues, however, there is a switch to a growth program that increases cell size only (hypertrophy). This increase in cell size often occurs via a variant endoreplication cycle that entails periodic genome duplication and continued cell growth without division, which generates increasingly large and polyploid cells [1,2]. One common polyploid cycle is called the endocycle, which is a repeated G / S cycle that completely skips mitosis [1–3]. The switch from mitotic cycles to endoreplication cycles is part of the normal growth program of specific tissues in a wide variety of organisms including humans [1,4].

There is a growing appreciation that cells can also switch to polyploid cycles in response to conditional inputs [1,2,5]. We have called these induced endoreplicating cells (iECs) to distinguish them from the programmed developmental endoreplicating cells (devECs) that occur during normal tissue growth [2,6]. Current evidence indicates that iECs can have beneficial effects on tissue homeostasis. Unscheduled iECs and cell fusion result in polyploid cells at wound sites, a process called wound induced polyploidy (WIP), which is beneficial for wound healing from insects to mammals [5,7–12]. After more significant tissue loss, an increase in iEC size can also regenerate tissue mass in a process called compensatory cellular hypertrophy (CCH) [1,5,7,10,13–18]. In contrast, other evidence indicates that in some contexts polyploidy can restrict regeneration [1,5,19–24]. Recently, there has been an intense interest in the contribution of iECs to cancer therapy resistance and cancer relapse. Human cancer cells switch to growth through unscheduled endoreplication cycles in response to various stresses and therapies, which generate therapy-resistant Polyploid Giant Cancer Cells (PGCCs) [2,19,25]. Some of these persistent PGCCs can then return to an error-prone division, which generates aneuploid cells that contribute to cancer progression [2,19,26]. Despite recent advances, much remains unknown about what regulates iEC growth and what determines their beneficial or detrimental effects on tissues and tumors *in vivo*.

*Drosophila* has been an important model system for investigating unscheduled endoreplication cycles *in vivo*. Work in *Drosophila* led to the discovery of WIP, CCH, and other iEC properties [2,5,8,14]. Moreover, like human cancers, several reports have documented polyploid cells in fly tumors [27–30]. One advantage of the *Drosophila* model is that specific cells can be experimentally induced to switch from mitotic cycles to unscheduled endoreplication cycles by inhibiting mitosis [2,6,31–39]. This approach permits determination of properties conferred to cells and tissues by a switch to endoreplication independent of other cellular and genetic variables during wound healing and tumorigenesis. We had previously shown that iECs were capable of returning to an error-prone division with high rates of chromosome instability (CIN) [6,40,41]. We recently found that unscheduled endocycles perturb the morphology of the somatic follicular epithelium that surrounds developing oocytes and result in pleiotropic defects in oogenesis [42]. These findings in *Drosophila* have contributed to current models for how transient endoreplication cycles of human PGCCs contribute to cancer therapy resistance and progression [2,19].

In this study, we address the impact of unscheduled endocycles on tissue growth using the *Drosophila* larval wing disc as a model, the anlage of the adult wing and thorax. The wing disc

is a premier developmental model system that has revealed fundamental principles of tissue patterning, growth, regeneration, and tumorigenesis [28,43–49]. We find that cell growth of iECs is limited and collectively results in tissue undergrowth. We also find that iECs have endogenous DNA damage and an activated Jun N-Terminal Kinase (JNK) signaling pathway but are apoptosis-resistant and instead enter a JNK-dependent senescent-like arrest. We uncovered that these arrested iECs maintain their position in the epithelium, unlike other growth-limited cells, and induce division of neighboring diploid cells. This compensatory pro-liferation, however, is incapable of rescuing normal tissue growth. Altogether, our findings indicate that unscheduled endocycles have unique attributes that result in tissue undergrowth and malformation, with important broader relevance to understanding their contribution to regeneration and tumorigenesis.

## Results

### iEC growth by hypertrophy slows over time

To evaluate the impact of unscheduled endocycles on tissue growth, we have utilized the *Drosophila* wing disc as a model. The two wing discs begin as outpouchings and by the end of embryogenesis are each composed of approximately 30 cells [50]. During larval and early pre-pupal stages, wing disc cells proliferate via canonical mitotic division cycles, which results in a final cell number of ~33,000 [49]. Cell fate and axial pattering of the wing disc is also progres-sively determined during larval life [49]. During metamorphosis, the wing disc undergoes morphological rearrangements and differentiates to form the adult wing blade, wing hinge, and most of the thorax [49].

We first addressed whether growth through an increase in cell size during induced endo-cycles can substitute for their normal growth through an increase in cell number by mitotic divisions. To do this, we created marked clones of iECs or control mitotic cycling cells in larval wing discs and measured their growth and DNA content from 24 to 72 hours by quantitative confocal microscopy (Fig 1A). Specifically, we used the GAL4 FLP-Out system to express *UAS-RFP* in clones with or without expression of the Cyclin A inhibitor Roughex (*UAS-rux*), which has been used previously to induce cells to switch to endocycles [39,51–54]. As expected, control mitotic clones increased in cell number over time at a rate that was consistent with pre-vious reports (Fig 1B and 1C) [55–57]. In contrast, clones continuously expressing *UAS-rux* were composed of fewer but larger cells (Fig 1B–1D). The majority of these *UAS-rux* clones were composed of single large, mononucleate cells, suggesting that they had switched to endo-cycles without dividing after induction, whereas others were composed of two or more cells because they had divided one or more times before switching to induced endocycles (Fig 1B and 1C). Unlike the diploid controls, the average DNA content (DAPI fluorescence) and cell size of the *UAS-rux* clones increased over time, consistent with whole genome duplications (WGD) and growth through cell hypertrophy during repeated G / S endocycles (Fig 1D and 1E). Although these iECs were larger, imaging of their dimensions in the x-z axis indicated that they did not protrude above or below their diploid neighbors in the epithelium (S1A-B" Fig). We could, therefore, use clone cross-sectional area to compare the total growth (accumu-lation of biomass) of iEC clones to that of control mitotic clones. This analysis showed that the growth of iEC and control clones were not significantly different during the first 48 hours (Fig 1F). By the 72 hour time point, however, the size of iEC clones was significantly smaller than that of mitotic controls (~2.3 fold) (Fig 1F). These results suggest that iECs can initially grow in cell size at a rate that is proportional to control clones that grow through an increase in cell number, but then iEC cell growth slows and lags behind that of mitotically dividing controls.

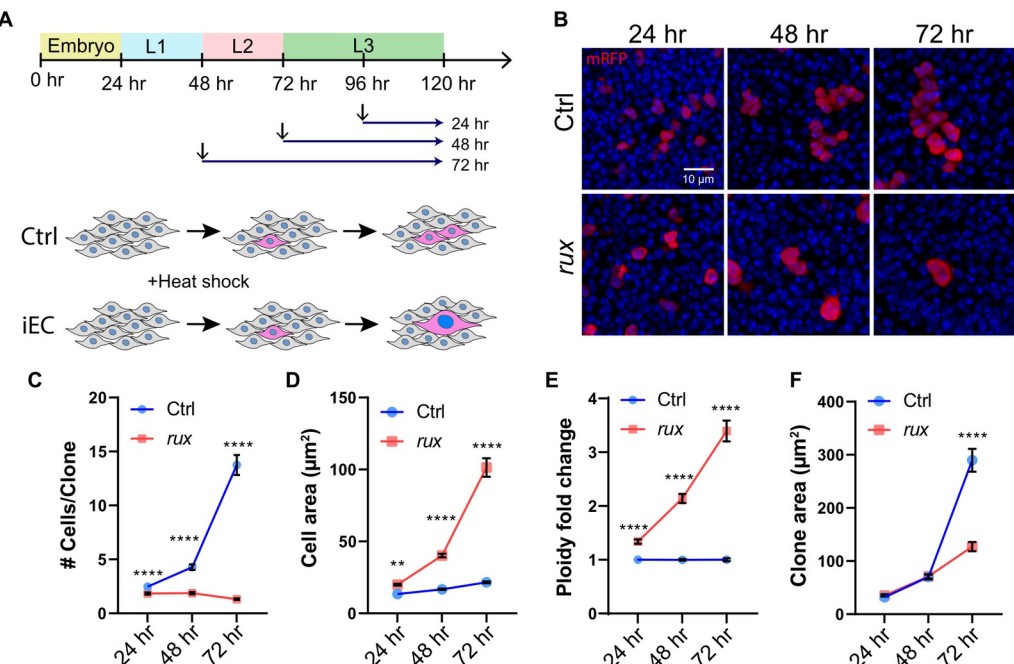

**Fig 1. Growth by iEC hypertrophy lags behind that from cell division.** (A) Experimental timeline: FLP-Out Clones of mitotic or endocycling cells were heat induced at different time points of larval development (arrows) and wing discs were dissected from late wandering 3rd (L3). (B) Examples of RFP-marked mitotic control (Ctrl) and *UAS-rux* expressing iEC (*rux*) clones at different times after induction. Scale bar = 10 μm (C-F) Mitotic control (Ctrl) and iEC (*rux*) FLP-Out clones were quantified for cell number per clone (C), average cell size per clone (D), fold increase in DNA content relative to diploid (E), and total clone area (F) over a period of 72 hours. ** p<0.01 **** p<0.0001.

## iECs contribute collectively to tissue undergrowth

The clonal analysis indicated that iEC growth slowed over time. It remained unclear, however, how tissue growth would be affected when larger groups of cells switch to unscheduled endo-cycles. To address this question, we used *en-GAL4* to express *UAS-rux* in the entire posterior compartment of the wing disc, with a temperature sensitive repressor of GAL4, GAL80ts, to control the duration of *en-GAL4* expression (Fig 2A) [58,59]. We induced iECs by shifting lar-vae from 18°C to 29°C at different developmental times and then dissected wing discs from wandering 3rd instar larvae (Fig 2A). Labeling of cells in S phase by incorporation of the nucle-otide analog 5-ethynyl 2′-deoxyuridine (EdU) together with antibodies against the mitotic marker phospho-Histone H3 (pH3) showed that anterior compartment cells continued to mitotically cycle whereas most *UAS-rux* expressing cells in the posterior compartment had switched to a G / S endocycle within 24 hours (S2A-B″″ Fig). To assess effects on tissue growth, we measured the size of the posterior compartment of the wing disc, marked by expression of *UAS-RFP*, as a normalized ratio to the total disc size. During the first 48 hours, the growth of the posterior compartment in *UAS-rux* discs was similar to that in control discs expressing only *UAS-RFP* (Fig 2B and 2C). By 72 hours, however, the posterior compartments of *UAS-rux* expressing wing discs were significantly smaller than control discs, a difference in size that was even greater after endocycling for 96 hours (Fig 2B and 2C). Consistent with the results of the clonal analysis, these results suggest that the collective growth of iEC by hypertrophy is comparable to that of diploid cells by mitotic divisions for the first 48 hours. However, beyond this period, iEC growth falls behind, resulting in significant tissue undergrowth.

We wished to determine how a switch to unscheduled endocycles affected the final develop-ment of the adult wing. However, *en-GAL4; UAS-rux* animals died as pupae preventing this

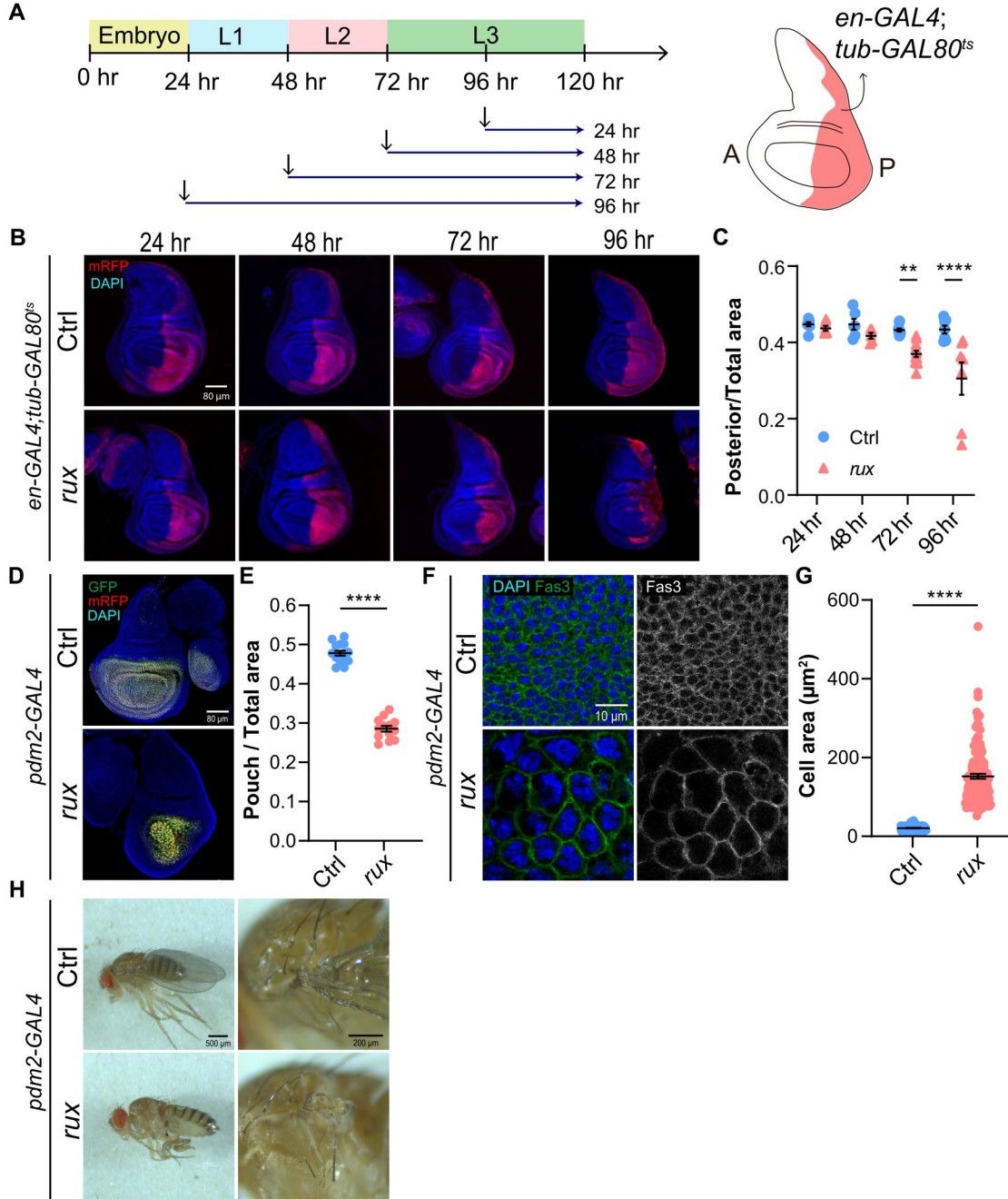

**Fig 2. Unscheduled endocycles cause wing disc and adult wing undergrowth.** (A) Experimental timeline: *en-GAL4* activity was induced in the posterior compartment of wing discs at different times by shifting from 18°C to 29°C (arrows), the GAL80ts nonpermissive temperature, followed by dissection during late wandering 3rd instar (L3). (B) Wing discs from control (Ctrl) and *UAS-rux* expressing (*rux*) wing discs after different times of induction. The posterior compartment is marked by *UAS-RFP* expression. Scale bar = 80 μm (C) Quantification of the ratio of posterior compartment to total wing disc area in control (Ctrl) or *UAS-rux* larvae (*rux*). (D-E) Induction of unscheduled endocycles in the wing pouch results in undergrowth. (D) *pdm2-GAL4* G-TRACE wing disc without (Ctrl) or with *UAS-rux* (*rux*) expression. Scale bar = 80 μm. (E) Quantification of the ratio of wing pouch area to total wing disc area during late 3rd instar for *pdm2-GAL4* alone (Ctrl) or *pdm-GAL4 UAS-rux* (*rux*). (F) *pdm2-GAL4* wing disc without (Ctrl) or with *UAS-rux* (*rux*) expression. Cell boundaries were labeled with anti-Fas3 antibody. Scale bar = 10 μm. (G) Quantification of the cross-sectional cell area for the two genotypes shown in F. (H) Two magnifications of adult fly wing phenotypes from control (Ctrl) and *pdm2-GAL4; UAS-rux (rux)* as shown in D. Scale bar = 500 μm and 200 μm. ** p<0.01 **** p<0.0001.

analysis. We therefore screened for other GAL4 drivers with wing expression that were adult viable with *UAS-rux*. One of these drivers was *pdm2-GAL4* [60]. Lineage analysis with the GAL4 Technique for Real-time And Clonal Expression (G-TRACE) showed that *pdm2-GAL4* was mostly expressed in the wing pouch, the central disc region fated to become adult wing blade (Fig 2D) [61]. Similar to our findings for cells in the posterior compartment, driving *UAS-rux* expression with *pdm2-GAL4* resulted in larger polyploid cells and a severe undergrowth of the wing disc pouch (Fig 2D and 2E). Quantification of cell size indicated that the average cross-sectional area of iEC was approximately 7 times larger than control diploid cells in the wing pouch, but with a notable wide variation of increase in iEC size (Fig 2F and 2G). Given that iECs and columnar control cells were the same height (average 35 μm), these measurements indicate the iEC were on average approximately 7 fold increased in cell volume [49]. Consistent with specific undergrowth of the total pouch area, *pdm2-GAL4 UAS-rux* adults were missing most or all of the wing blade but had a normal wing hinge and notum (Fig 2H). These results indicate that an unscheduled switch to endocycles results in undergrowth of wing disc tissue and failure to form a normal adult wing.

## Endocycling cell death does not contribute significantly to tissue undergrowth

Tissue growth is a balance between cell proliferation and cell death [48]. We therefore examined the extent to which the death of iECs was contributing to tissue undergrowth. Labeling *tub-GAL80^ts^ en-GAL4 UAS-rux* for the activated form of the caspase Dcp-1 revealed that apoptosis of posterior compartment iECs occurred at a low frequency that was comparable to that of diploid cells in the anterior (Fig 3A–3C) [62]. Blocking iEC apoptosis by expressing the caspase inhibitor *UAS-p35* did not significantly change the undergrowth of the posterior compartment (Fig 3D–3J). Together, these results suggest that the low frequency of iEC death does not make a major contribution to tissue undergrowth.

We had previously shown that both developmental and induced endocycling cells in other tissues repress the apoptotic response to replication stress or DNA damage induced by ionizing radiation (IR) [6,34,40,41]. To determine if wing disc iECs also repress this apoptotic response, we induced iEC in the posterior compartment with *tub-GAL80^ts^ en-GAL4 UAS-rux* for 3 days, irradiated with 4,000 rads of gamma rays, and four hours later labeled with antibodies against activated Dcp-1. IR resulted in numerous Dcp-1 labeled diploid cells in the anterior compartment but not endocycling cells in the posterior compartment of the same disc (Fig 3K–3M). These results indicate that wing disc iECs are refractory to apoptosis after induction of high levels of genotoxic stress.

## Heterogeneity of growth among endocycling cells causes tissue undergrowth

The data indicated that the hypertrophic growth of unscheduled endocycling cells is limited and collectively compromises tissue growth. To investigate the growth dynamics of individual iECs, we quantified the progressive doubling of iEC DNA content during repeated endocycle S phases by measuring total DAPI intensity. We induced endocycles in the posterior compartment and at different times thereafter labeled cells in S phase by incubating wing discs *in vitro* with the nucleotide analog EdU for one hour. The S phase fraction and DNA content of posterior pouch iECs were normalized to that of control, mitotic cycling cells in the anterior pouch of the same wing disc. Induction of endocycles resulted in a progressive increase in average DNA content per cell over time, with an ~10-fold average increase over diploid by 96 hours (~32C average DNA content) (Fig 4A–4D' and 4E). Notably, however, the variance in DNA

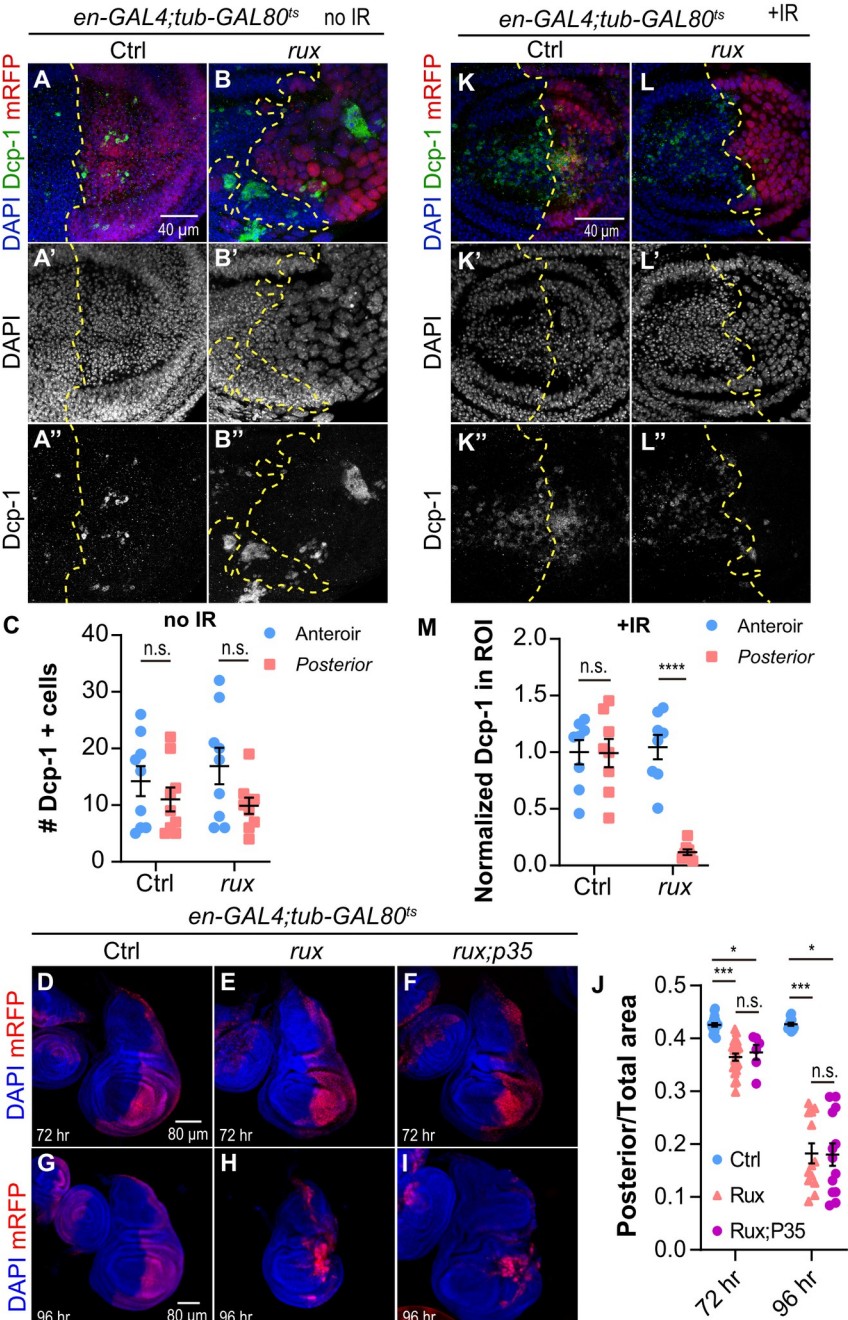

**Fig 3. Death of iECs does not play a major role in tissue undergrowth.** (A-B''') Wing discs without (ctrl, A-A'') or with *UAS-rux* (rux, B-B''') expression in the posterior compartment for three days and labeled with anti-activated caspase Dcp-1 antibody. Scale bar = 40 μm. (C) Quantification of Dcp-1+ cells in the anterior or posterior wing disc for the genotypes shown in A-B''. (D-I) Wing disc with expression of *UAS-mRFP* only (D,G, Ctrl), *UAS-mRFP UAS-rux* (E,H, *rux*), or *UAS-mRFP UAS-rux* and the caspase inhibitor *UAS-p35* (F, I, *rux; p35*) in the posterior compartment for 72 hours (D-F) or 96 hours (G-I). Scale bar = 80 μm. (J) Quantification of the ratio of posterior compartment to total wing disc area for the three genotypes shown in D-I. (K-L'') iECs are apoptosis resistant. Wing discs without (Ctrl, K-K'') or with *UAS-rux* (*rux*, L-L'') expression in the mRFP-marked posterior compartment for three days were irradiated (IR+) with gamma-rays and labeled with anti-activated caspase Dcp-1 antibody four hours later. Scale bar = 40 μm. (M) Quantification of anti-Dcp-1+ fluorescent volume in the same size region of interest (ROI) in anterior or posterior wing disc for the genotypes shown in K-L''. **** p<0.0001, *** p<0.001, n.s. not significant.

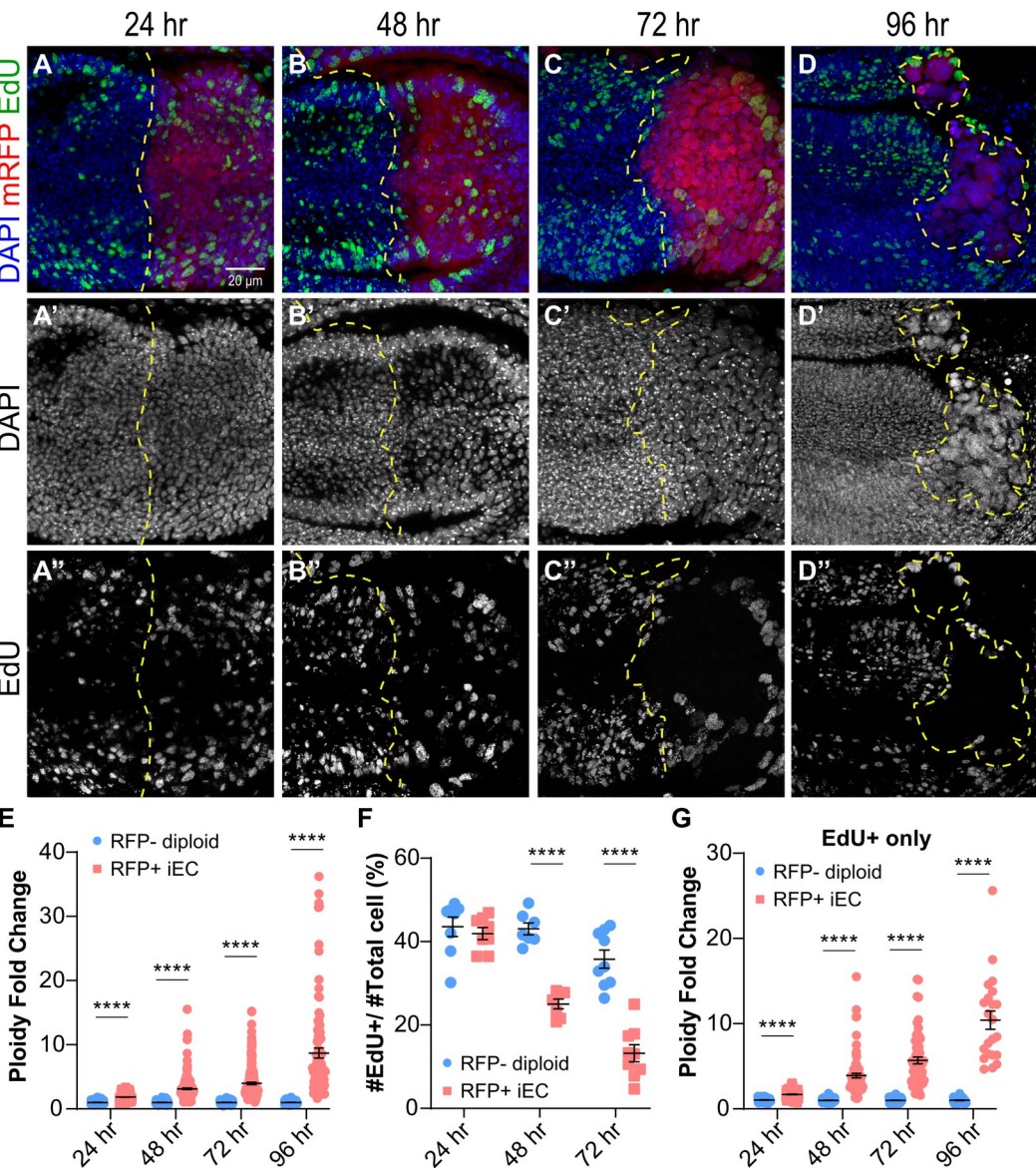

**Fig 4. Arrest of iECs at different terminal ploidies causes tissue undergrowth.** (A-D") 3rd instar wing discs with *UAS-rux and UAS-RFP* expression in the posterior for the indicated durations of time were dissected and incubated in EdU *in vitro* for one hour. Scale bar = 20 μm. (E) Fold change in DNA content (DAPI fluorescence) of mRFP-positive posterior iECs relative to diploid mRFP-negative anterior diploid cells from the same wing discs after different durations of *en-GAL4* expression. (F) Fraction of EdU positive cells in the anterior and posterior compartments of wing discs after the indicated durations of *en-GAL4* expression. (G) Ploidy fold change of EdU positive cells only. **** p<0.0001.

content among iECs also progressively increased over time, with some cells remaining at lower ploidy while others continued to increase in ploidy (at 96 hours, a range of 4C-111C, or ~1–7 endocycles) (Fig 4E). Over the same time points, the fraction of EdU-labeled iECs in S phase progressively decreased (Fig 4A–4D" and 4F). The progressive decrease in S phase fraction together with the increase in the variance of DNA content suggested that iECs were arresting at different terminal ploidies. To further address this, we compared the ploidy of the EdU labeled iECs to that of all iECs. This revealed that in later time points the EdU labeled cells were biased towards those with higher ploidy, suggesting that those cells with lower ploidy had

been arrested earlier (Fig 4G). The variance in iEC DNA content is consistent with the data that there is a large variance in their cell size (Fig 2F–2G). The iECs that continued to endocycle replicated their DNA at the normal rate as indicated by the constant ratio of EdU to DAPI fluorescence (S3A Fig). Despite prevalent cell growth arrest at 72 hours, iECs continued to maintain their position to form a continuous epithelium (S3B-E" Fig). Altogether, these results suggest that there is significant heterogeneity in growth among iECs, with some arresting while others continuing to cycle, thereby resulting in different terminal ploidies that collectively contribute to tissue undergrowth.

## iECs undergo a senescent-like arrest and have DNA damage near heterochromatin

The evidence indicated that iECs were arrested after different numbers of endocycles, but the nature of the arrest was unclear. A clue came from metanalysis of our RNA-Seq data of S2 cells in culture that had been induced into endocycles by Cyclin A RNAi [32]. This analysis revealed that these S2 iECs have increased expression of many genes whose orthologs are associated with a senescent arrest of human and mouse cells (121 of 177 fly genes increased with FDR p <0.05) (S1 Table) [63]. Labeling of wing discs indicated iECs also express the senescent associated genes Matrix metalloproteinase 1 (Mmp1) and senescent-associated beta galactosidase (SA-β-GAL) (Fig 5A–5D'). In addition, iECs induced by overexpressing *fizzy-related (fzr)* also had high levels of SA-β-GAL, indicating that senescence is not specific to *rux* expression (S4A-C' Fig). These results suggested that growth of wing disc iECs is limited by a senescent-like arrest *in vivo*.

It is known that senescence can be activated in response to a variety of different stresses, with a common one being genotoxic stress [64–68]. Antibody labeling against the phosphorylated form of H2Av ($\gamma$H2Av) showed the iEC had elevated numbers of DNA repair foci relative to control diploid cells in the same wing disc (Fig 5E–5F'). Double labeling for $\gamma$H2Av and the heterochromatic marker H3K9me3 indicated that these repair foci were enriched near and within nuclear heterochromatic domains (Fig 5G–5H'). These results suggest that iECs have replication stress and DNA damage in difficult-to-replicate heterochromatic DNA, similar to our previous findings for devECs [41]. To investigate whether DNA damage induces an iEC growth arrest, we increased DNA damage with irradiation and determined if it enhanced tissue undergrowth. iECs were induced in the posterior wing compartment of *en-GAL4; UAS-rux* larvae for 24 hours before irradiation with 4,000 rads of gamma rays, followed by 48 hours of recovery before dissection and measurement of the ratio of posterior / total wing disc area (IR 48 hr, S5A Fig). The undergrowth of the posterior compartment was not significantly different between control and irradiated wing discs, suggesting that increased DNA damage is not sufficient to enhance tissue undergrowth (S5B–S5D Fig).

Further analysis of these iECs 4 hours and 48 hours after IR revealed that despite having extensive DNA damage, they did not have active caspase (S5E–S5H" Fig). Imaging these discs in the y-z axis indicated that dying anterior cells extruded from the disc epithelium, whereas the posterior iECs did not delaminate from the disc epithelium (S5I–S5J Fig). Altogether, these results suggest that iECs have endogenous DNA damage, but are apoptotic resistant and instead undergo a senescent-like arrest while maintaining their position in the epithelium.

## An activated JNK pathway in iECs induces a senescence-like arrest

We next asked what could be mediating the iEC arrest. One pathway that is known to play a major role in stress response entails activation of the Jun N-terminal Kinase (JNK, a.k.a *Drosophila basket*, *bsk*) [69–72]. It is known that activation of the JNK signaling pathway can

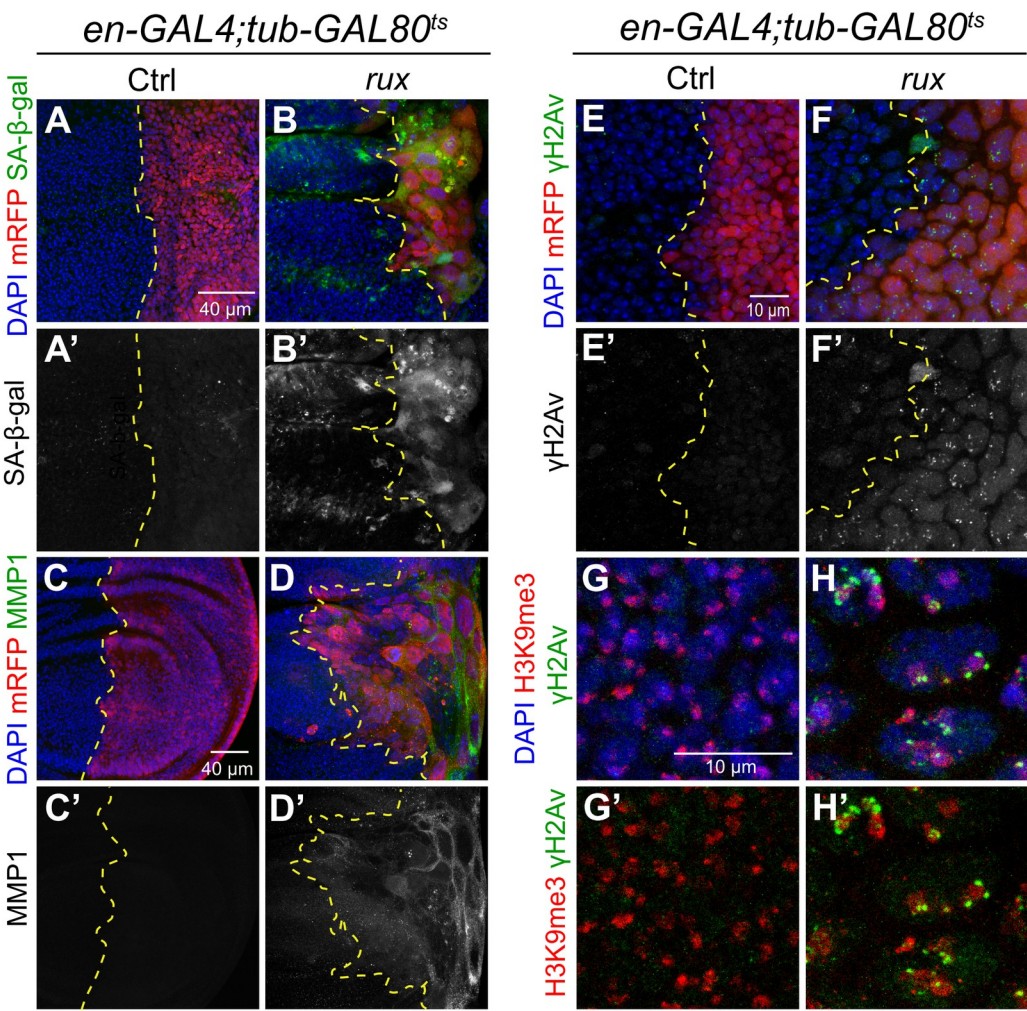

**Fig 5. iECs express senescence-associated genes and have DNA damage.** (A-D) iECs have increased SA-β-Gal activity (A-B') and MMP1 (C-D') expression after 72 hours induction in posterior wing disc compartment. Scale bar = 40 μm. (E-F') DNA damage foci in posterior iECs labeled with anti-γH2Av antibody. Scale bar = 10 μm. (G-H') iECs have DNA damage near heterochromatin. High magnification of iECs labeled with antibodies against damage marker γH2Av and heterochromatin marker H3K9me3. Scale bar = 10 μm.

induce a cell cycle arrest and the expression of MMP1 and other genes that were upregulated in iECs [69,73–75]. To address whether iECs have elevated JNK activity, we analyzed the expression of the JNK activity reporter, TRE-GFP [76]. TRE-GFP expression was significantly increased in iECs indicating that they have elevated JNK activity (Fig 6A–6B'). To address whether JNK activity is required for iECs senescence, we inhibited JNK activity using a GAL4-inducible, dominant negative form of *bsk (UAS-bsk^DN)*. Expression of *bsk^DN* in iEC greatly reduced expression of TRE-GFP and SA-β-Gal, indicating that JNK activity is required for SA-β-Gal expression in iECs (Fig 6A–6C'). The p53 tumor suppressor is also known to play a major role in the senescent response to a variety of stresses in flies and mammals [75,77]. To test if p53 mediates the senescent response in iECs, we inhibited p53 activity with a dominant negative form of p53 (*UAS-p53^R155H*). While in control experiments the expression of *p53^R155H* strongly inhibited the apoptotic response of diploid cells to IR, it had no effect on SA-β-Gal expression in iECs (S6A–S6D' Fig). These results indicate that JNK signaling, but not p53, is required to induce expression of SA-β-Gal in iECs.

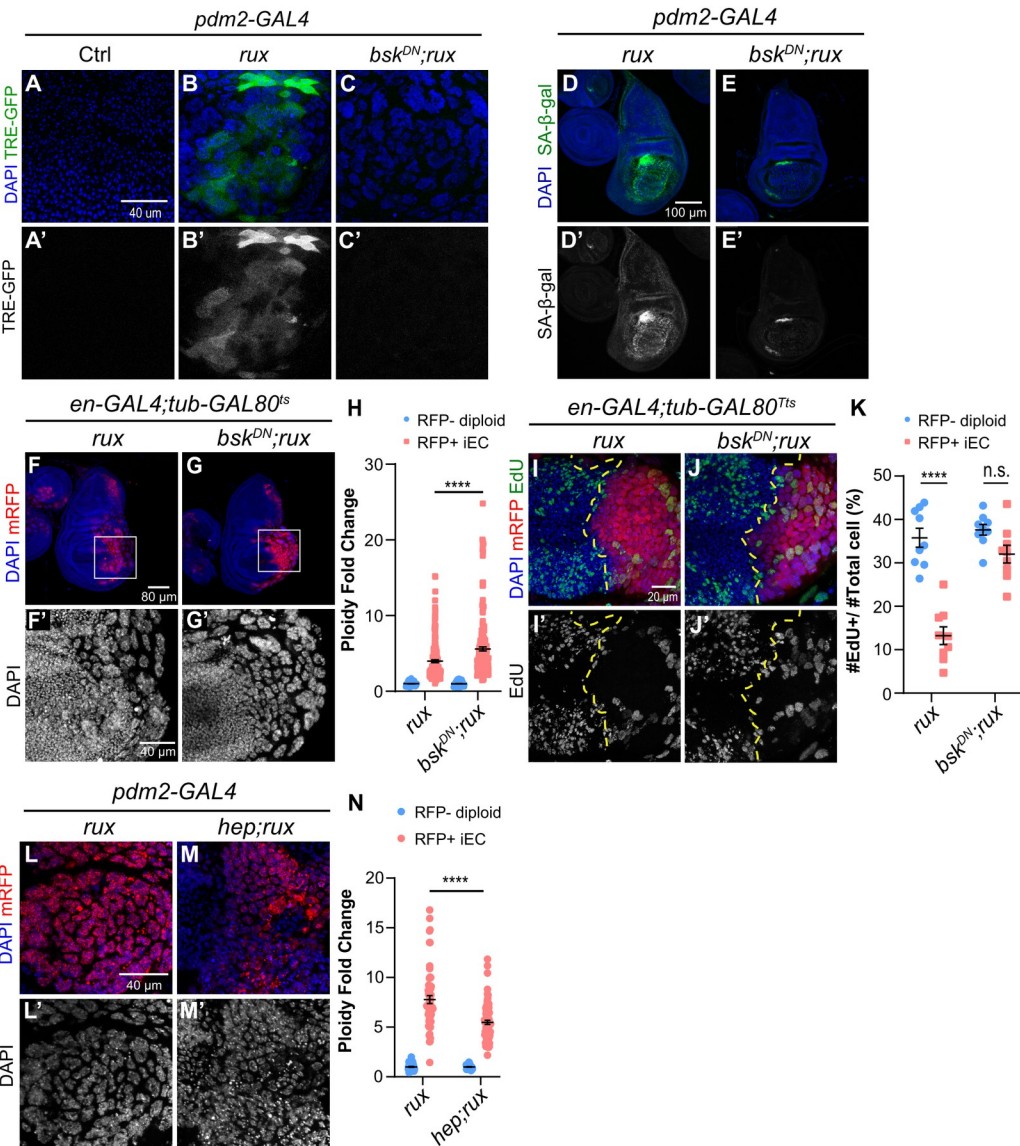

**Fig 6. JNK signaling negatively regulates iEC growth.** (A-C') Expression of the JNK reporter, TRE-GFP, in wing discs with *pdm2-GAL4* only (A, A'), or with *UAS-rux* (B, B'), or *UAS-rux* and *UAS-bsk^DN* (C, C'). Scale bar: 40 μm. (D-E') SA-β-Gal activity in wing discs expressing *UAS-rux* alone (D-D') or *UAS-rux* and *UAS-bsk^DN* (E-E') in the wing pouch. Scale bar: 100 μm. (F-G') Wing discs expressing *UAS-rux* alone (F-F') or *UAS-rux* and *UAS-bsk^DN* (G-G') in the posterior compartment for 72 hours. Scale bar: 40 μm. (H) Quantification of DNA content for genotypes shown in F-G'. The DNA content of RFP+ iEC in the posterior was normalized to the average DNA content of anterior RFP-negative diploid cells. The dataset used for *UAS-rux* (rux) expression group is the same as the 72 hours induction in Fig 4E. (I-J') 3rd instar wing discs with *UAS-rux* alone (I-I') or *UAS-rux* and *UAS-bsk^DN* (J-J') expression in the posterior for 72 hours and were labeled with EdU *in vitro* for an hour after dissection. Scale bar: 20 μm. (K) Quantification of S phase fraction in the anterior and posterior wing disc compartments for genotypes shown in I-J'. The dataset used for *UAS-rux* (rux) expression group is the same as the 72 hours induction in Fig 4F. (L-M') Wing discs expressing *UAS-rux* alone (L-L') or *UAS-rux* and *UAS-hep* (M-M') in the wing pouch. Scale bar: 40 μm. (N) Quantification of DNA content for genotypes shown in L-M'. DNA content of RFP+ iEC in the wing pouch was normalized to the average DNA content of RFP-negative diploid cells. **** p<0.0001, n.s. not significant.

To address whether JNK activity restrains iEC cycling and growth, we measured polyploid DNA content in *en-GAL4; UAS-rux* iEC with and without expression of *UAS-bsk^DN*. Inhibition of JNK activity resulted in larger and more polyploid iECs (Fig 6F–6H). Some of these

larger iECs had distinct polytene chromosomes that had foci of $\gamma$H2Av labeling near heterochromatin, suggesting that replication stress in these cells is not downstream of JNK signaling (S6E–S6H"" Fig). Consistent with this increase in DNA ploidy, inhibition of JNK activity also rescued the fraction of EdU-labeled iECs in S phase to levels that were comparable to that of the control mitotic cycling cells in the anterior compartment, suggesting that JNK is required for endocycle arrest (Fig 6I–6K). We also performed the opposite experiment and increased JNK signaling in iECs by overexpressing *hemipterous* (*hep*), the upstream kinase that phosphorylates and activates JNK [78,79]. Expression of *UAS-hep* restricted iEC growth and resulted in smaller and less polyploid iECs (Fig 6L–6N). Expression of an activated *hep (hep^{CA})* in larval salivary glands also inhibited developmental endocycles (S6A–S6D Fig). To determine if p53 also restrains iEC growth, we expressed the dominant negative *UAS-p53^{R155H}* in iECs, which did not increase their polyploid DNA content nor the fraction of EdU-labeled cells (S7E–S7G Fig). These results suggest that the activation of JNK pathway, but not p53, restrains iEC growth.

## The iEC senescence-like arrest is reversible

Although senescent arrest was once thought to be permanent, it is now clear that some types of senescence are reversible with cells able to return to division cycles [80,81]. To test if the iEC senescence-like arrest is reversible, we conducted temperature shift experiments to induce and then subsequently repress *UAS-rux* expression. *pdm2-GAL4 tub-GAL80^{ts} / UAS-rux* larvae were shifted from 18°C to 29°C to induce endocycles in the wing pouch for four days, and then switched back to 18°C to turn off *UAS-rux* expression. Control *pdm2-GAL4 / UAS-rux* larvae without *tub-GAL80^{ts}* subjected to the same temperature shifts had continuous *UAS-rux* expression. After one day of recovery at 18°C, wing discs were labeled for SA-β-GAL activity and the mitotic chromosome marker anti-pH3. Control discs with continuous *UAS-rux* expression had SA-β-GAL activity in the large, polyploid iEC with no evidence of anti-pH3 labeling in the wing pouch (Fig 7A–7A'). In contrast, after one day of GAL80 repression of *UAS-rux* expression, wing discs had sporadic large cells with condensed, pH3-labeled chromosomes in various stages of segregation (Fig 7B–7C'). Many of these polyploid divisions had multiple, aberrant chromatin masses and fragmentation (Fig 7C–7C'). Labeling the cell periphery and mitotic spindle showed that many of these aberrant divisions are multi-polar (Fig 7D–7E'). SA-β-GAL activity perdured in these cells resulting in double labeling for this senescent marker and pH3, further indicating that they had been in a senescent arrest (Fig 7C). After three days of recovery from *UAS-rux* expression, some cells in the wing pouch labeled with anti-PH3 and had lower, but variable, SA-β-GAL activity (Fig 7F–7I'). A lineage analysis over three days using the G-TRACE system confirmed that some iEC daughter cells can continue to divide (Fig 7J–7L'''). These results suggest that the iEC senescence-like arrest is reversible and that at least some cells can return to error-prone division cycles.

## Growth-compromised iECs induce compensatory proliferation of diploid neighbor cells

Wing discs can regenerate after physical wounding by inducing proliferation of the remaining cells to replace missing tissue, a process known as compensatory proliferation [48,82–84]. Growth-compromised and dying cells also induce proliferation of healthy adjacent cells to ensure proper disc growth during development [44, 85–89]. This includes "undead" cells that have an active apoptotic pathway that is blocked downstream by expression of caspase inhibitors [48,74,75,87]. We therefore wondered whether growth-compromised and apoptotic-resistant iECs would induce proliferation of neighboring cells. Labeling of *en-GAL4 UAS-rux* wing discs with EdU revealed that there was an increase in proliferation of diploid cells in the

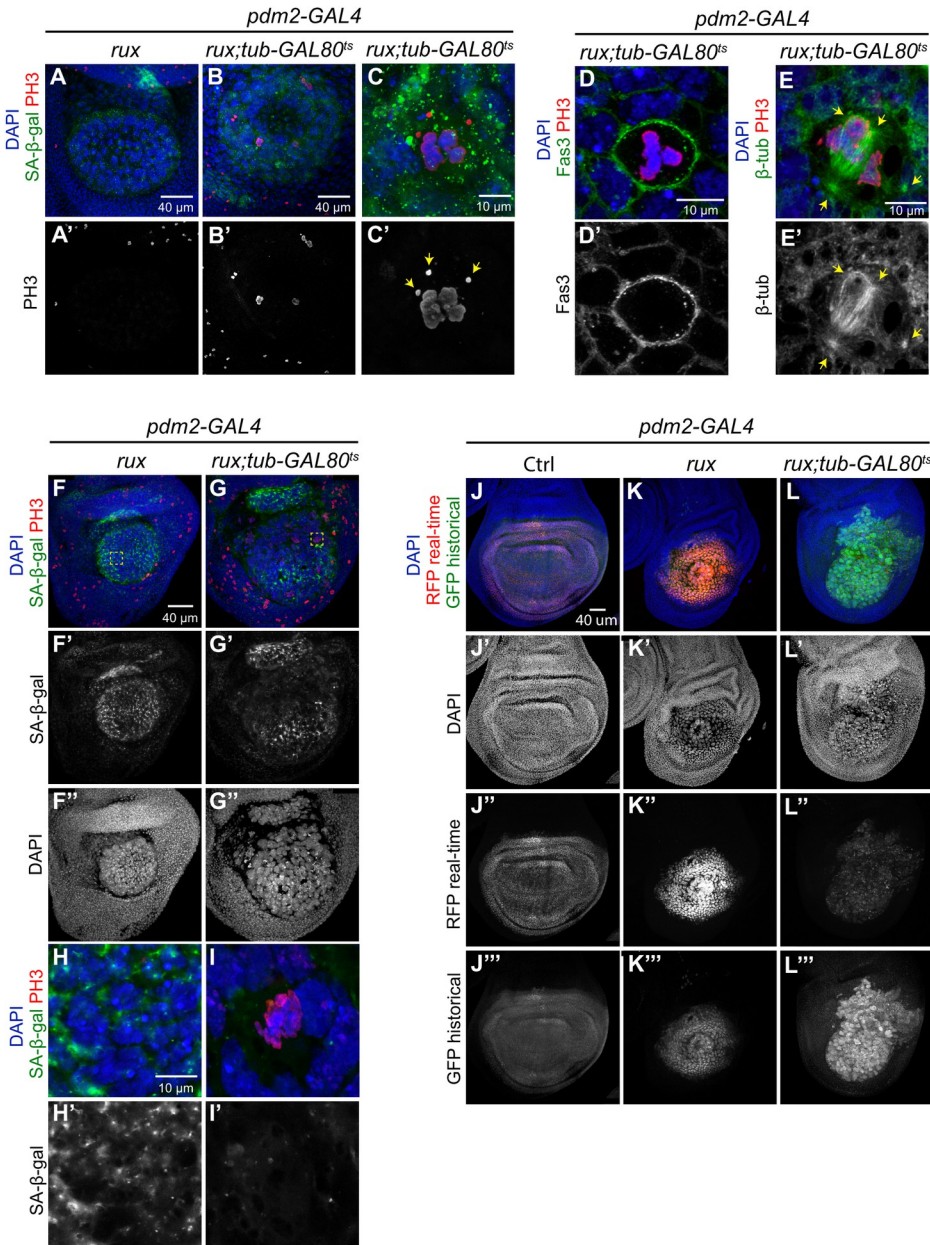

**Fig 7. The iEC senescent-like arrest is reversible.** (A-C') *Pdm2*-GAL4 wing discs expressing *UAS-rux* alone (A-A') or *UAS-rux* and *tub-Gal80^{ts}* (B-C') were raised at 29˚C for 96 hours and shifted to 18˚C for 24 hours recovery. Cells were co-labeled with mitotic marker anti-pH3 antibody and senescence marker SA-β-Gal. Arrows in C' indicate chromatin fragments off of the metaphase plate in a polyploid iEC mitosis. Scale bars: 40 μm (A-B') and 10 μm (C-C'). (D-E') Dividing iEC co-labeled with mitotic chromosome marker anti-pH3 and cell periphery marker anti-Fas3 (D-D') or anti-pH3 and mitotic spindle marker anti-β-tubulin (E-E'). Arrows in E-E' indicates microtubule organizing center. Scale bar: 10 μm. (F-I') Wing discs expressing *UAS-rux* alone (F-F", H-H') or *UAS-rux* and *tub-Gal80^{ts}* (G-G", I-I') were raised at 29˚C for 96 hours and shifted to 18˚C for 72 hours recovery. Cells were co-labeled with mitotic marker anti-pH3 antibody and senescence marker SA-β-Gal. (H-I') Higher magnifications of boxed area in panels F and G. Scale bar: 40 μm (F-G") and 10 μm (H-I'). (J-L'") Lineage analysis indicates iEC daughter cells continue to proliferate. (J-J'") Control pdm2-GAL4 G-TRACE wing disc with real time GAL4 expression (RFP) and historical GAL4 expression (GFP) in the wing pouch. (K-L'") pdm2-GAL4 G-TRACE wing disc with *UAS-rux* alone (K-K'") or *UAS-rux* and *tub-Gal80^{ts}* (L-L'") were raised at 29˚C for 96 hours and shifted to 18˚C for 72 hours recovery. Scale bar: 40 μm.

regions adjacent to iECs after 96 hours induction relative to controls (Fig 8A–8D). There was also a reduction in EdU labeling in the anterior compartment distant from the A-P boundary, consistent with previous evidence that activation of DILP-8 can repress cell division far from the tissue insult [90–92]. We also occasionally observed hyperplastic overgrowth of the notum, but this phenotype was not highly penetrant (Fig 8B–8B"). Despite the increased proliferation of diploid neighbors, 3rd instar wing discs were severely undergrown and malformed (Fig 8B–8B"). iEC continued to express *enGAL4 UAS-RFP*, indicating they maintained their posterior identity, whereas proliferating cells around them labeled for Ci, a marker of anterior compartment identity (Fig 8E–8F") [93]. This observation is consistent with previous evidence that the A-P compartment lineage restriction can be violated during regeneration [94,95]. These results suggest that iECs induce compensatory proliferation, but that this increased proliferation cannot fully compensate for iEC growth defects to generate a normal wing disc.

## Discussion

Induced endocycling cells can have either beneficial or pathological effects, but much remains unknown about their impact on tissue growth. We have addressed this question by using a system to induce endocycles in the *Drosophila* wing disc. Our findings reveal that iECs are like developmental endocycling cells in that they have persistent DNA damage near heterochromatin and a repressed apoptotic response to genotoxic stress [6,40,41]. Unlike devECs, however, iECs were limited in their hypertrophic cell growth and arrested after different numbers of endocycles, which resulted in tissue undergrowth and malformation. iEC growth arrest was senescent-like and dependent on activation of the JNK pathway, but was reversible, with some cells having the capacity to return to an error-prone division. We also found that iEC undergrowth induced the proliferation of neighboring diploid cells, but this compensatory proliferation failed to regenerate a normal wing disc and adult wing. Altogether, our data reveal that unscheduled endocycles have unique attributes that can have severe deleterious consequences for tissue growth (Fig 8G). These findings have broader implications for understanding how unscheduled iECs promote wound healing or aberrant growth in development and cancer.

The regulation of cell size and the coordinate scaling of cellular pathways and compartments is an active area of research [96–98]. For cells in the mitotic cell cycle, size homeostasis is determined by the rate at which a cell grows (accumulation of mass) balanced by the frequency with which it divides to halve cell volume [98–101]. We uncoupled this process by inducing cells to skip mitotic cell division, which engaged an oscillating G / S endocycle that progressively increased cell size and DNA content. The subsequent heterogenous arrest of iEC growth contrasts with *Drosophila* devECs which can grow to very large cell size and DNA content through repeated G / S endocycles (for example salivary glands ~1130 $\mu m^2$ and >1,000C) [1,102,103]. The large size of some devECs indicates that iECs are not arresting because of a fundamental upper limit on cell size. Why then are iECs growth limited? Activation of the JNK stress pathway in iECs suggests that they differ from devECs by having a distinct stress or stress response that triggers their growth arrest. We found, however, that overexpression of constitutively active JNK Kinase in salivary glands can inhibit developmental endocycles. iECs did have DNA damage near heterochromatin, suggestive of replication stress, but this property is shared with devECs [41]. It is known that an uncoordinated increase in the size of diploid cells can result in cytoplasmic dilution and a senescent arrest, which raises whether a failure of scaling in iECs limits their growth [104–109]. It is also possible that active developmental remodeling of checkpoint and metabolic pathways in devECs supports their growth to large cell size. An important future goal is to determine how iECs and devECs differ to reveal mechanisms of normal and aberrant growth.

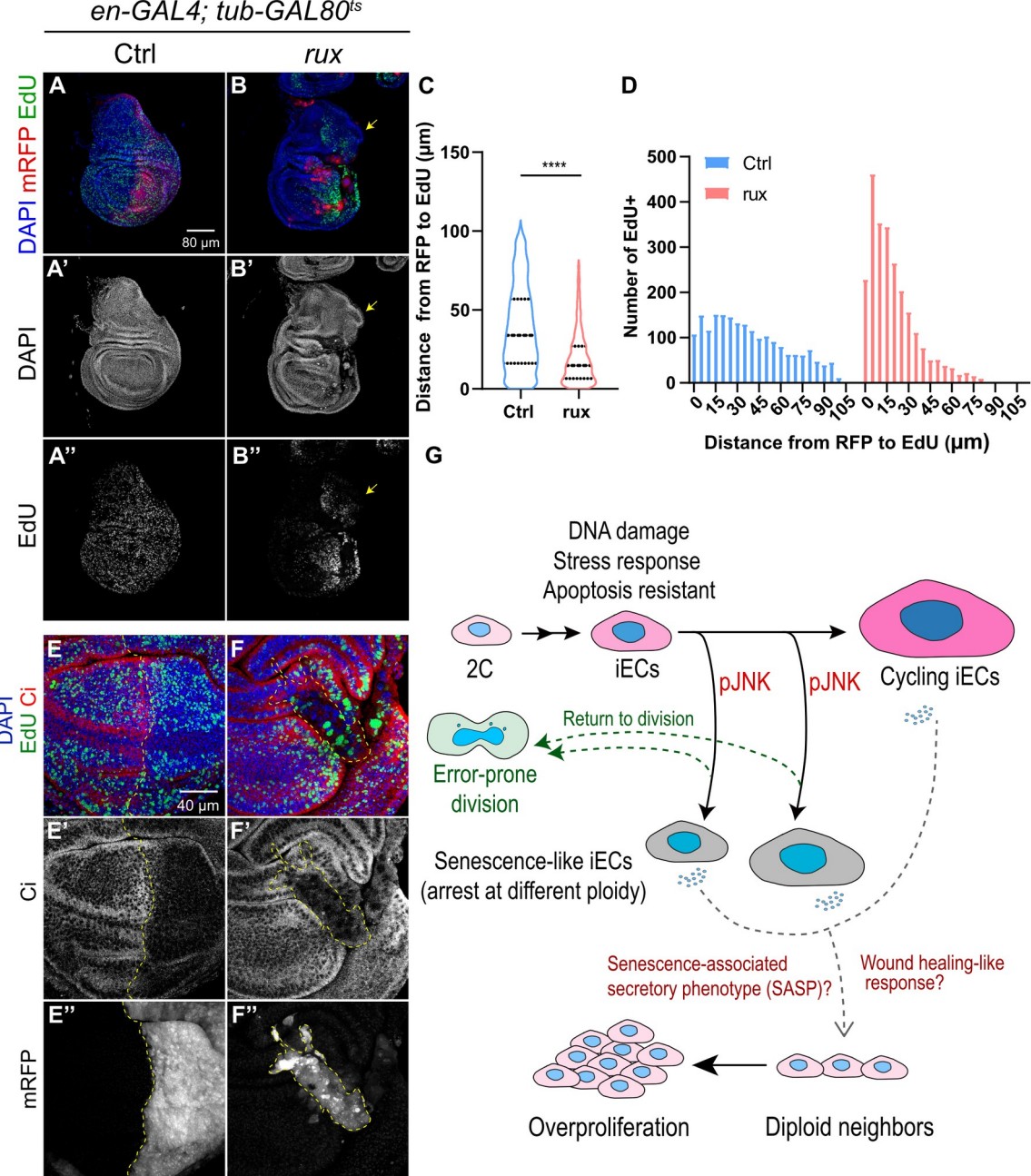

**Fig 8. iECs induce proliferation of neighboring diploid cells.** (A-B") 3<sup>rd</sup> instar wing discs expressing RFP alone (Ctrl, A-A") or with 96 hours of *UAS-rux* expression (rux, B-B") in the posterior compartment were labeled with EdU *in vitro* for one hour. Arrows in B-B" point to hyperplastic overgrowth of the notum. Scale bar: 80 μm. (C-D) Distribution of distances between RFP and EdU (C) and histogram of numbers of EdU+ cells at different distances from the RFP+ posterior for the genotypes shown in A-B". (E-F") 3rd instar wing discs expressing RFP alone (Ctrl, E-E") or with 96 hours of UAS-rux expression (rux, F-F") in the posterior were incubated with EdU *in vitro* for one hour and co-labeled with anterior marker anti-Ci and posterior marker anti-RFP. Scale bar: 40 μm. (G) Summary and model for a reversible senescent-like and JNK-dependent growth arrest of iECs and their effect on neighboring cells. See text for details.

We found that S2 iEC *in vitro* had increased expression of many genes whose orthologs are upregulated in human and mouse senescent cells. Wing disc iECs *in vivo* also expressed senescent-associated genes MMP1 and SA-β-Gal, common markers for senescence in flies and mammals [63,110–112]. iECs had other properties of senescent-like cells, including elevated DNA damage, apoptotic resistance, JNK activation, and cell cycle arrest [110,113–116]. What defines a senescent state, however, is a topic of active discussion. Current evidence suggests that there is considerable heterogeneity in the character of a senescent-like arrest among different cell types and in response to different stresses [110,114,117–119]. Consistent with that emerging concept, our data indicate that, unlike other types of senescent cells, p53 is not required for iEC senescent-associated gene expression and arrest [75,77]. Moreover, our finding that some senescent iECs were capable of returning to division is consistent with the current view that some types of senescence are reversible [80,81,114,120]. To fully understand the impact of iECs on tissue and tumor growth, therefore, it will be important to further determine how the iEC senescent-like arrest compares to other types of senescent arrest, which will also contribute to ongoing efforts in the field to define the diversity in the senescent response [121].

Our data indicate that iECs have properties that are both similar to and different from the response of imaginal disc cells to other challenges (Fig 8G). Like iECs, it has been documented that diploid disc cells induce JNK in response to injury and a variety of other stresses, and that these cells can induce proliferation of neighboring cells to regulate tissue growth and regeneration [48,72,73,75,122,123]. One common outcome of JNK activation is apoptosis and delamination of the cell from the epithelium [48,75,124]. Blocking apoptosis in these cells by expressing the viral caspase inhibitor p35 results in "undead" cells that are biased towards a JNK-dependent senescent arrest, which delaminate and can induce neoplasia of neighboring tissue [48,75,115,116,125,126]. Most relevant to our study, it has been reported that forcing errors in mitosis generates aneuploid wing disc cells that activate JNK and apoptose, but if apoptosis is blocked these cells senesce, delaminate, migrate, and induce neoplasia [74,127–129]. In contrast, we found that polyploid iECs were naturally undead and senescent-prone and often maintained their position in the epithelium with only rare delamination. Moreover, although iECs induced proliferation of neighbors and occasionally caused local overgrowth, we did not observe frequent neoplasia. Thus, unscheduled endocycles that reduplicate the genome elicit a cell fate and tissue response that differs from that of aneuploid cells with unbalanced genomes. Defining what governs these alternative polyploid and aneuploid cell fates will be important for understanding how they affect tissue regeneration and tumorigenesis.

Our results have important relevance to iECs in wound healing and regeneration. The ability of wound induced polyploidy (WIP) to promote wound healing and compensatory cellular hypertrophy (CCH) to regenerate lost post-mitotic tissue is conserved from flies to mammals [5,10,13,14,130]. It was shown that even in the absence of cell cycling the continued growth of G2 arrested wing disc cells can generate a wing disc compartment of normal size [131]. Most relevant to our study, Weigmann and colleagues reported that induction of endocycles in the anterior compartment of the wing disc can, in some cases, generate a normal sized compartment and disc [37]. These and other studies suggest that an increase in cell size can rescue tissue growth in the absence of cell division, which is seemingly at odds with our findings that iEC arrest causes tissue undergrowth. Most previous studies, however, required only modest hypertrophy to rescue tissue mass. For example, in the study by Weigmann and colleagues, the ability of CCH hypertrophic growth to regenerate tissue was evaluated for only two days after endocycle induction, which is consistent with our findings that hypertrophy of iECs can initially compensate for the absence of cell division for the first two days of growth [37]. Our evidence that JNK eventually restricts iEC growth is also consistent with previous evidence that

JNK activation restrains the growth of unscheduled endocycling cells at wound sites in *Drosophila*, and further suggests that this may be an intrinsic property of induced endocycles [9]. Moreover, in mice and humans, iECs can repair damaged kidney tubules through CCH, but these iECs subsequently senesce and contribute to kidney fibrosis and cancer [18,132–135]. Altogether, these observations suggest that there may be limits to iEC cell growth that can lead to tissue malformations and neoplasia.

Our findings are also relevant to the contribution of unscheduled endocycling cells to cancer (Fig 8G). Polyploid giant cancer cells (PGCCs) are like our iECs in that they grow via endoreplication cycles, are resistant to cell death, and can return to divisions that generate aneuploid daughter cells [2,19]. Moreover, recent evidence suggests that PGCCs senesce and produce cytokines that stimulate proliferation of neighboring cells [136–138]. Our similar findings for experimentally-induced endocycling cells suggest these properties are inherent to an unscheduled endocycle state that is independent of other genetic and cellular complexities of human tumors. The similarity to iECs at wound sites suggests that unscheduled endocycles are another example that supports the perspective that tumors are like "wounds that do not heal" [139,140].

## Materials and methods

### *Drosophila* melanogaster Stocks

Information about fly strains, genetics, and other information was obtained from FlyBase [141]. Fly strains were raised in standard Bloomington *Drosophila* stock center media (https://bdsc.indiana.edu/information/recipes/bloomfood.html) at 25˚C. For FLP-Out clonal experiments, larvae were raised at 25˚C before and after 30 min heat-shock at 37˚C. For GAL80<sup>ts</sup> experiments, larvae were raised at 18˚C then shifted to 29˚C to activate GAL4 drivers. For return to division experiments in Fig 7, larvae were shifted back to 18˚C for one to three days before analysis. Fly strains that were obtained from Bloomington *Drosophila* Stock Center (BDSC, Bloomington, IN, USA) are listed in S2 Table. Strains that were used in individual panels are listed in S3 Table.

### Immunofluorescence microscopy and quantification

Late 3rd instar larvae were dissected in either PBS (phosphate buffered saline) or Grace's solution, fixed in 6% formaldehyde, permeabilized with PBT (phosphate buffered saline with 0.1% Triton X-100), and blocked in 5% Normal Goat Serum as previously described [40]. The antibodies and software used for this study are listed in S2 Table. The following antibodies were used: RFP (Takara, Cat#632496) at 1:1000, p-Tyrosine (Millipore Sigma, Cat#05-321X) at 1:500, GFP (Invitrogen, Cat#A11122) at 1:1000, MMP1 (Developmental Studies Hybridoma Bank, Cat#3A6B4, 3B8D12, 5H7B11) at 1:20 for each, γH2Av (Developmental Studies Hybridoma Bank, Cat#UNC93-5.2.1) at 1:1000, H3K9me3 (active motif, Cat#39162) at 1:1000, pH3 (Cell signaling, Cat#9706S) at 1:1000, cleaved *Drosophila* Dcp-1 (Cell Signaling, Cat#9578 S) at 1:200, Fasciclin III (Developmental Studies Hybridoma Bank, Cat#7G10) at 1:100, beta-Tubulin (Developmental Studies Hybridoma Bank, Cat#E7) at 1:50. For the EdU labeling, larvae were dissected and incubated in 10 μM EdU in Grace's medium for 1 hr and processed according to the manufacturer's instructions (Invitrogen, Cat#C10337). The tissues were stained with DAPI (0.5 μg/ml), imaged on a Leica SP8 confocal, and quantified using Image J. Dcp-1+ fluorescent volume of confocal z stacks in Fig 3M was quantified by using Imaris with the setting of 200 pixel in width and 350 pixel in height as the region of interest (ROI). The distance between RFP+ area to EdU+ cells in Fig 7C–7D was quantified using Imaris.

## Senescence gene metanalysis

For evaluation of senescent gene expression, we used the list of senescent genes upregulated in human and mouse cells that was published by Saul et al. [63]. We mapped the *Drosophila* orthologs and paralogs corresponding to human genes on that list using DRSC integrated ortholog prediction tool (DIOPT) [142]. This resulted in a list of 177 *Drosophila* genes for which we had RNAseq data from CycA RNAi iEC in S2 cells [32] (S1 Table). The iEC RNA-Seq data is available at NCBI-GEO under accessions GSE121955.

## SA-β-GAL activity labeling

The larvae were dissected in PBS, fixed in 4% formaldehyde, washed by PBS with 2% BSA, and incubated at 37˚C for 3 hours according to the manufacturer's instructions (Invitrogen, Cat#C10850). The tissues were stained with DAPI (0.5 μg/ml) and imaged on a Leica SP8 confocal.

## γ-Irradiation

Flies were irradiated with a total of 40 Gy (4000 rads) from a cesium source, and 48 hours later labeled with anti-cleaved *Drosophila* Dcp-1 antibody (Cell Signaling, Cat#9578S), and anti-γH2Av (UNC93-5.2.1, Developmental Studies Hybridoma Bank) as previously described [143].

## Statistical analysis

Statistical analysis was performed using GraphPad Prism. All statistical graphs were shown as mean ± S.E.M. from at least three biological replicates. Two-way ANOVA was used for Figs 1C–1F, 2C, 3C, 3M, 4F, 6K, S5D, and S7F. The Mann-Witney test was performed for Figs 2G, 4E, 4G, 6H, 6N, 8C and S7G. Welch's *t* test was performed for Fig 2E. One-way ANOVA was used for Fig 3J. Primary data for all quantifications are available in S4 Table.

## Supporting information

**S1 Fig. Transverse section of FLP-Out clones.** A transverse confocal section of the wing pouch epithelium in the x-z plane showing mRFP-marked mitotic (Ctrl, A-A") or *UAS-rux* expressing clones (rux, B-B"). FLP-Out clones were heat induced and allowed to grow for 72 hr. Image axis for each panel: anterior (left) to posterior (right); apical (up) to basal (down). Scale bar: 10 μm.
(TIF)

**S2 Fig. Induction of endocycles in the posterior compartment of the wing disc.** (A-B"")
Wing disc with *en-GAL4* driven expression of either *UAS-mRFP* only (Ctrl, A-A"") or *UAS-mRFP and UAS-rux* (*rux*, B-B"") in the posterior compartment after 24 hours of *en-GAL4* expression. 3rd instar wing discs were incubated in EdU *in vitro* for an hour, followed by detection of EdU incorporation and labeling with anti-pH3 antibody to detect cells in S and M phases respectively. Scale bar: 80 μm.
(TIF)

**S3 Fig. Unscheduled endocycles have a normal DNA synthesis rate and maintain position in the epithelium.** (A) DNA synthesis rate is similar between iEC and diploid cells. 3rd instar wing discs with *UAS-rux and UAS-RFP* expression in the posterior for the indicated durations of time were dissected and incubated in EdU *in vitro* for one hour. DNA synthesis rate was quantified by the ratio of EdU intensity to DAPI intensity per nucleus. Shown are data for

mRFP-positive posterior iECs relative to mRFP-negative anterior diploid cells from the same wing discs. (B-E") Transverse sections across the wing pouch in the x-z plane (B-C") or y-z plane of posterior compartment (D-E"). Shown are *en-GAL4* discs expressing *UAS-mRFP* only (Ctrl, B-B", D-D") or *UAS-mRFP and UAS-rux* (*rux*, C-C", E-E") in the posterior compartment after 72 hours of *en-GAL4* expression. Image axes: B-C"—anterior (left, RFP-) to posterior (right, RFP+); apical (up) to basal (down). D-E"—dorsal (left) to ventral (right); apical (up) to basal (down). Scale bar: 10 μm.
(TIF)

**S4 Fig. Fzr over-expression induces iECs that express SA-β-Gal.** (A-C') *Pdm2*-GAL4 wing discs express *Pdm2*-GAL4 only (Ctrl, A-A'), *UAS-rux* (*rux*, B-B') or *UAS-fzr* (*fzr*, C-C') were collected at 3rd instar and labeled with senescence marker SA-β-Gal. Scale bar: 100 μm.
(TIF)

**S5 Fig. Radiation DNA damage does not enhance undergrowth nor induce apoptosis of iEC.** (A) Experimental timeline: *en-GAL4* activity was induced in the posterior compartment of wing discs by shifting from 18˚C to 29˚C (arrow), the GAL80ts nonpermissive temperature, and collected at 3rd instar. Larvae were irradiated either 48 hours (IR 48 hr) or 4 hours (IR 4 hr) before dissection. (B-C) Examples of wing discs expressing *UAS-mRFP* only (Ctrl, B) or UAS-*mRFP* and *UAS-rux* (*rux*, C) in the posterior compartment 48 hours after irradiation. Scale bar: 100 μm. (D) Quantification of the ratio of posterior compartment to total wing disc area for genotypes shown in B-C. n.s. not significant. (E-H") IR induces DNA damage but not iEC apoptosis. iECs were induced for 72 hours and were irradiated either 48 hours (IR 48 hr, E-F") or 4 hours before dissection (IR 4 hr, G-H"). Irradiated wing discs were labeled with anti-γH2Av antibody for DNA damage and anti-Dcp-1 antibody for caspase activity. Scale bar: 40 μm. (I, J) IR induces apoptosis and delamination of diploid cells but not iECs. A transverse x-z section through the wing pouch epithelium of an irradiated wing disc expressing *UAS-mRFP and UAS-rux* in the posterior compartment. The wing disc was irradiated 48 hours before dissection and cells were labeled with anti-Dcp-1 antibody to detect caspase activity. Yellow arrowheads in J point to position of basally extruding Dcp-1-positive diploid cells in the anterior compartment. Image axes: anterior (left, RFP-) to posterior (right, RFP+); apical (up) to basal (down). Scale bar: 10 μm.
(TIF)

**S6 Fig. JNK activation inhibits developmental and induced endocycles but DNA damage is JNK-independent.** (A-D) Salivary glands expressing *fkh-GAL4* alone (Ctrl, A, C) or with a constitutively active form of *hep*, *UAS-hep*CA (*hep*CA, B, D) were labeled with DAPI to assess nuclear DNA content. Yellow arrowheads in A, B point to salivary glands. Scale bars are 200 μm for A,B and 40 μm for C,D. (E-F') Images of iECs without (E, E'), or with (F, F') expression of *bsk*DN. Anterior diploid cells are on left and posterior iECs with polytene chromosomes are on right and marked with RFP. (G-H"") Images of single wing disc nuclei in posterior cells expressing either *UAS-rux* alone (G-G"") or *UAS-rux* with *UAS-bsk*DN (H-H"") for 120 hours. Polytene chromosomes are labeled with anti-γH2Av antibody for DNA damage and anti-H3K9me3 antibody for heterochromatin and counterstained with DAPI.
(TIF)

**S7 Fig. p53 is not required for the iEC senescence-like arrest.** (A-B') Dominant negative p53 inhibits apoptosis in diploid cells. Wing discs expressing *UAS-mRFP* only (ctrl, A-A') or *UAS-mRFP* and *UAS-p53*R155H (*p53*R155H, B-B') for 72 hours in the posterior compartment were irradiated (+IR) and then labeled with anti-Dcp-1 for caspase activity four hours later. Scale bar: 40 μm. (C-D') Dominant negative p53 does not inhibit SA-β-Gal expression in iEC.

SA-**β**-Gal activity in wing discs expressing either *UAS-rux* alone (*rux*, C-C') or *UAS-rux and UAS- p53^{R155H}* (*p53^{R155H}*, *rux*; D-D') in the wing pouch. Scale bar: 100 μm. (E-E') Dominant negative p53 does not alter iEC growth arrest. Wing discs expressing *UAS-rux* with *UAS-p53^{R155H}* were induced in the posterior compartment for 72 hours and were labeled with EdU *in vitro* for one hour. Scale bar: 40 μm. (F) Quantification of S phase fraction in the anterior and posterior wing disc compartments for genotypes shown in E-E'. The dataset used for *UAS-rux* (rux) expression group is the same as the 72 hours induction in Fig 4F. (G) Quantification of DNA content for genotypes shown in E-E'. DNA content of RFP+ posterior iEC in the wing pouch was normalized to the average DNA content of RFP-negative anterior diploid cells. The dataset used for *UAS-rux* (rux) expression group is the same as the 72 hours induction in Fig 4E. n.s. not significant.
(TIF)

**S1 Table. Differential senescence-associated genes expression in S2 iECs.**
(XLSX)

**S2 Table. Reagents and software used in this study.**
(XLSX)

**S3 Table. Fly strain used for this Study.**
(XLSX)

**S4 Table. Raw data for quantifications.**
(XLSX)

## Acknowledgments

We thank H. Herriage, P. Rangarajan for their feedback on the manuscript. We thank E.A. Bach and Bloomington Drosophila Stock Center for fly strains, J. Powers and A. Kun from IU Light Microscopy Imaging Center (LMIC) for imaging advice, and FlyBase for bioinformatic support.

## Author Contributions

**Conceptualization:** Yi-Ting Huang, Brian R. Calvi.

**Funding acquisition:** Brian R. Calvi.

**Investigation:** Yi-Ting Huang, Lauren L. Hesting.

**Methodology:** Yi-Ting Huang, Brian R. Calvi.

**Project administration:** Yi-Ting Huang.

**Resources:** Yi-Ting Huang.

**Supervision:** Brian R. Calvi.

**Validation:** Yi-Ting Huang, Lauren L. Hesting, Brian R. Calvi.

**Visualization:** Yi-Ting Huang.

**Writing – original draft:** Yi-Ting Huang, Lauren L. Hesting, Brian R. Calvi.

**Writing – review & editing:** Yi-Ting Huang, Lauren L. Hesting, Brian R. Calvi.

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
