## [Decision Letter · Decision Letter 0]

21 May 2024

Dear Dr Calvi,

Thank you very much for submitting your Research Article entitled 'An unscheduled switch to endocycles induces a reversible senescent arrest that impairs growth of the Drosophila wing disc' to PLOS Genetics.

The manuscript was fully evaluated at the editorial level and by independent peer reviewers. The reviewers appreciated the attention to an important topic but identified some concerns that we ask you address in a revised manuscript.

We therefore ask you to modify the manuscript according to the review recommendations. Your revisions should address the specific points made by each reviewer.

We hope to receive your revised manuscript within the next 60 days. If you anticipate any delay in its return, we would ask you to let us know the expected resubmission date by email to plosgenetics@plos.org.

Yours sincerely,

Hongyan Wang, Ph.D.

Academic Editor

PLOS Genetics

Fengwei Yu

Section Editor

PLOS Genetics

Reviewer's Responses to Questions

**Comments to the Authors:**

Reviewer #1: This study investigates how unscheduled polyploidy (iEC) affects cell and tissue growth using the Drosophila wing imaginal disc as a model. Overall, this is a timely, rigorous, and intriguing study! The authors utilize the fruit fly’s genetic and cell biological tool kit to conditionally induced polyploidy using a temperature sensitive Gal80 to regulate Gal4 dependent expression of the cyclin A inhibitor, rux, in a subset of disc epithelial cells. Interestingly, they observe that polyploidy via the endocycle is sufficient to compensate for tissue growth in the short-term, but prolonged polyploidization leads to a defect in the wing disc development. They go on to demonstrate that iEC cells exhibit similarities (genotoxic stress) and differences (senescence-like state) to programmed polyploidy. Their findings are remarkably similar to other models of iECs in mammals and Drosophila demonstrating that wing disc is a useful model to address questions on iEC’s role and regulation of tissue/organ physiology. Most intriguing is their observation that a subset of the iECs cease to cycle entering a senescent-like state marked by expression MMP1 and beta-gal. Their discovery that the senescent-like state is dependent on JNK signaling is of particular importance as this offers a novel target to test in mammalian systems, such as the kidney, where the persistence of tubule epithelial iECs has been shown to contribute to fibrous induced kidney disease.

Therefore, this study will be of wide interest to those working in the polyploidy field and beyond.

Major Concern:

Validate that the senescent-like state is not specific to the overexpression of Rux by inducing polyploidy via another genetic perturbation (i.e. Fizzy-related, CycE, or E2F1 RNAi). Then repeating the wing disc staining with MMP1 and beta-gal (Figure 5) would be sufficient to address this point.

Minor Concern:

None. A very well written manuscript which nicely puts the significance of the study in the context of the field.

Reviewer #2: Huang et al use the wing imaginal disc of Drosophila larvae as an experimental tissue to understand the consequences to tissue growth of inducing endoreplication in an otherwise population of proliferating diploid cells. They report that induced endocycling cells (iEC) arrest cell cycle progression at various states of ploidy because of activation of JNK signaling, and consequently are unable to restore full growth to the imaginal disc. This arrest has several features of senescent cells, although they can reenter mitosis when the iEC inducing signal is removed. The iEC have properties of developmentally controlled endocycling cells (devEC), most notably an increase in DNA damage within or near constitutive heterochromatin. The experiments in support of these observations are very well performed and nicely described. The major new advance is the description of how JNK signaling controls the behavior of iEC and the consequent effects on tissue growth, and this information should be helpful when thinking about pathological iEC as has been observed in several cancers. The comments below are intended to help the authors improve the manuscript.

1. Many readers, particular non-fly readers, likely won’t know what rux is or why it induces iEC. I suggest adding a sentence (e.g. page 5) and a citation(s) indicating that rux a cyclin A/cdk inhibitor and that loss of the activity of this kinase is a common way of inducing endoreplication.

2. Most of the experiments are well quantified, but several are not and it’s unclear why not. e.g. Figure 3A-G, Figure 7, and some others. Are some stainings not readily quantifiable? Probably should be consistent throughout the manuscript.

3. In Figure S4A there is no control showing how much damage was induced by the IR, making the negative result more difficult to interpret. In the subsequent panels these data are shown. Can we assume that 4000 rads always uniformly induces the same amount of damage in every disc? And why is there Dcp1 staining in the posterior half of the control disc AND the anterior half of the experimental disc in panels S4E’’’ and S4F’’’? Finally, why is it that iEC don’t activate caspase? Is p53 never activated?

4. The polytenization of the highly polyploid iEC after JNK inhibition is striking and very interesting. In addition to showing that JNK signaling is required for attenuation of tissue growth via senescence, it suggests that devEC don’t activate JNK; e.g. in the highly polyploid and polytene salivary glands. Is that known? What would happen if JNK was ectopically activated in the salivary gland? Would that block endoreplication and/or polytenization?

5. Figure 6O-Q: it’s hard to tell what’s really going on here. Can the spindle be labeled to better demonstrate multipolarity? Are those PH3 positive chromosome fragments all in the same iEC? A cell cortex marker might help in this regard.

6. The experiments in Figure 7 are the least well developed and thus the least convincing. In panel B there appears not to be EdU labeling in the anterior half of the disc, unlike control. Why is that? That’s not true for other discs shown in the manuscript. Also, the data are not quantified. In addition, why would you expect to regenerate a “normal wing” if there's already a bunch of polyploid iEC in the posterior compartment? The logic for this expectation doesn’t make sense. This line of experimentation seems a bit preliminary and could be removed.

Reviewer #3: Huang et al. report that cells induced to endocycle (iEC) in Drosophila wing discs arrest growth, become senescent-like, resist apoptosis, and stimulate their diploid neighbors to divide with the overall result of reduced tissue size. iEC cells activate JNK signaling, and this pathway was found to be responsible for cell cycle arrest/slowdown in iEC cells. These are previously unknown properties of iEC cells. Traditionally thought to be unique to certain cell types during insect and mammalian development, we now appreciate endocycles for their role in regeneration, carcinogenesis, and drug resistance of tumors. The insights reported by Huang and colleagues, therefore, should be of interest to the wide readership of PLoS Genetics. While I find most figures convincing and supportive of the author’s conclusions, there are two exceptions that should be addressed before publication.

1. Fig. 6O-Q’ is confusing. Based on the mitotic marker PH3, endocycling cells can return to a mitotic state once rux has been shut off, leading the authors to conclude that senescence in iECs is reversible. However, senescence marker SA-beta-gal is, if anything, higher after rux shut off. So, based on this marker, iEC cells are still in senescence. If the senescence is really reversible, shouldn’t SA-beta-gal also disappear? If it persists because it is a stable protein, the authors should monitor other senescence markers with shorter half-lives to test their idea more rigorously.

2. Fig. 7 is also confusing and does not appear to show what the authors describe. We are supposed to see increased proliferation (EdU) of diploid cells that surround iEC cells, but the most striking results seems to be the reduction of EdU to near absence in the diploid cells of the anterior compartment. The RFP+ posterior compartment appears highly fragmented so are the ‘diploid neighbors’ posterior cells that somehow did not express rux? Or are they anterior cells that are now next to posterior cells? Overall, the possibility that iEC cells induce non-autonomous proliferation is better addressed using well-isolated iEC clones as in Fig. 1 where the boundary between iEC and diploid neighbors are clearly defined.

This is a minor concern but the effect of p35 co-expression may be better tested in the background where rux expression results in a greater reduction in growth than what is shown in Fig. 3F (for example, the 96 hr condition in Fig. 2C).

For improved presentation, please label the panels with the name of GAL4 driver in Fig. 2 (en or pdm2). And clarify whether the published RNAseq dataset from S2 cells induced to endocycle (re-analyzed in supplemental Table 1) are from cells induced with cyclin A dsRNA or Myb dsRNA (since both were used to induce endocycles in the prior publication).

**Have all data underlying the figures and results presented in the manuscript been provided?**

Reviewer #1: Yes

Reviewer #2: Yes

Reviewer #3: Yes

PLOS authors have the option to publish the peer review history of their article (what does this mean?). If published, this will include your full peer review and any attached files.

Reviewer #1: No

Reviewer #2: No

Reviewer #3: No

---

## [Decision Letter · Decision Letter 1]

6 Aug 2024

Dear Dr Calvi,

We are pleased to inform you that your manuscript entitled "An unscheduled switch to endocycles induces a reversible senescent arrest that impairs growth of the Drosophila wing disc" has been editorially accepted for publication in PLOS Genetics. Congratulations!

Yours sincerely,

Hongyan Wang, Ph.D.

Academic Editor

PLOS Genetics

Fengwei Yu

Section Editor

PLOS Genetics

Comments from the reviewers (if applicable):

Reviewer's Responses to Questions

**Comments to the Authors:**

Reviewer #1: The authors addressed my concern that expression of the senescent marker, bgal, is dependent on iEC and not the genetic means to induce them. Thus, I have no further concerns and recommend the manuscript be accepted for publication.

Reviewer #2: Thank you for clearly and thoroughly addressing all of my comments and questions.

Reviewer #3: Huang et al. report that cells induced to endocycle (iEC) in Drosophila wing discs arrest growth, become senescent-like, resist apoptosis, and stimulate their diploid neighbors to divide with the overall result of reduced tissue size. iEC cells activate JNK signaling, and this pathway was found to be responsible for cell cycle arrest/slowdown in iEC cells. These are previously unknown properties of iEC cells. Traditionally thought to be unique to certain cell types during insect and mammalian development, we now appreciate endocycles for their role in regeneration, carcinogenesis, and drug resistance of tumors. The insights reported by Huang and colleagues, therefore, should be of interest to the wide readership of PLoS Genetics. The authors have addressed my concerns in the revised version and I recommend the publication of this work.

**Have all data underlying the figures and results presented in the manuscript been provided?**

Reviewer #1: Yes

Reviewer #2: Yes

Reviewer #3: Yes

PLOS authors have the option to publish the peer review history of their article (what does this mean?). If published, this will include your full peer review and any attached files.

Reviewer #1: No

Reviewer #2: No

Reviewer #3: No

**Data Deposition**

http://datadryad.org/submit?journalID=pgenetics&manu=PGENETICS-D-24-00361R1

**Press Queries**

---

## [Editor Report · Acceptance letter]

26 Aug 2024

PGENETICS-D-24-00361R1 

An unscheduled switch to endocycles induces a reversible senescent arrest that impairs growth of the Drosophila wing disc 

Dear Dr Calvi, 

We are pleased to inform you that your manuscript entitled "An unscheduled switch to endocycles induces a reversible senescent arrest that impairs growth of the Drosophila wing disc" has been formally accepted for publication in PLOS Genetics! Your manuscript is now with our production department and you will be notified of the publication date in due course.

With kind regards,

Anita Estes

PLOS Genetics

On behalf of:
